# How Does Hedonic Aroma Impact Long-Term Anxiety, Depression, and Quality of Life in Women with Breast Cancer? A Cross-Lagged Panel Model Analysis

**DOI:** 10.3390/ijerph19159260

**Published:** 2022-07-28

**Authors:** Marta Pereira, Célia Sofia Moreira, Pawel Izdebski, Alberto C. P. Dias, Cristina Nogueira-Silva, M. Graça Pereira

**Affiliations:** 1Psychology Research Centre (CIPsi), School of Psychology, University of Minho, 4704-553 Braga, Portugal; martinha.marquespereira@gmail.com; 2Faculty of Sciences, Center of Mathematics (FCUP & CMUP), University of Porto, 4169-007 Porto, Portugal; celiasofiamoreira@gmail.com; 3Institute of Psychology of the Kazimierz Wielki, University in Bydgoszcz, 85-064 Bydgoszcz, Poland; pawel@ukw.edu.pl; 4Centre of Molecular and Environmental Biology (CBMA), University of Minho, 4710-057 Braga, Portugal; albertocpdias66@gmail.com; 5Centre of Biological Engineering (CEB), University of Minho, 4710-057 Braga, Portugal; 6Life and Health Sciences Research Institute, School of Medicine, University of Minho, 4710-057 Braga, Portugal; cristinasilva@med.uminho.pt; 7ICVS/3B’s—PT Government Associate Laboratory, 4806-909 Guimarães, Portugal; 8Department of Obstetrics and Gynaecology, Hospital de Braga, 4710-243 Braga, Portugal

**Keywords:** breast cancer, chemotherapy treatment, anxiety, depression, quality of life

## Abstract

Depression and anxiety are common symptoms during and after adjuvant chemotherapy treatment for breast cancer (BC), with implications on quality of life (QoL). The present study evaluates the temporal relationship between anxiety, depression, and QoL (primary outcomes), as well as the impact of hedonic aroma (essential oils) on this relationship. This is a secondary analysis of a previously reported randomized controlled trial, with two groups: an experimental group (n = 56), who were subjected to the inhalation of a self-selected essential oil during chemotherapy, and a control group (n = 56), who were only subjected to the standard treatment. The hedonic aroma intervention occurred in the second (T1), third (T2), and fourth (T3) chemotherapy sessions, three weeks apart from each other. The follow-up (T4) assessments took place three months after the end of the treatment. Cross-lagged panel models were estimated in the path analysis framework, using structural equation modeling methodology. Regarding the control group, the cross-lagged panel model showed that anxiety at T1 predicted anxiety at T3, which in turn predicted both QoL and depression at T4. In the experimental group, hedonic aroma intervention was associated with stability of anxiety and QoL over time from T1 to T3, with no longitudinal prediction at T4. For women undergoing standard chemotherapy treatment, anxiety was the main longitudinal precursor to depression and QoL three months after chemotherapy. Thus, essential oils could complement chemotherapy treatment for early-stage BC as a way to improve long-term emotional and QoL-related adjustment.

## 1. Introduction

Breast cancer (BC), is the leading cancer in women worldwide [1] (Ferlay et al., 2019). Tragically, research has shown an increase in incidence in developed countries [2,3,4], especially in high purchasing power parity countries [5]. BC is a heterogeneous disease that entails different treatments, namely systemic therapy (chemotherapy, hormone therapy, and immunotherapy) and local therapy (surgery and radiotherapy) [6]. This diversity of treatments allows for a more personalized response to tumor characteristics, improving patient survival outcomes [6]. The increase in survival rate [2,4] is associated with several long-term effects on patient’s quality of life (QoL) due to the side effects of treatment [6]. For this reason, currently, the focus is not only on short-term adverse effects, but also on physical and psychological sequelae that can persist for a long time [6].

Throughout the disease trajectory, the chemotherapy treatment (CT) is the most feared period for women with BC [7], mainly due to its side effects [8]. Thus, CT is interpreted as a source of stress, which manifests in increased levels of anxiety and depression [9,10]. According to Lazarus and Folkman’s [11] transactional model of stress and coping, stressful experiences are understood as transactions that occur between a person and the environment. When environmental stimuli are perceived (primary appraisal) as threatening, challenging, or harmful (i.e., stressful), individuals evaluate (secondary appraisal), the resources available to alleviate the effects of these stimuli. If environmental demands exceed personal resources, well-being is affected. Thus, this cognitive appraisal and its coping style will have an impact on immediate outcomes (such as emotional and physiological changes), which extend over time, and may impact health outcomes, such as psychological functioning and QoL.

The prevalence of depression and anxiety symptoms in women with BC is high, both during [9,10] and after treatment [12]. More specifically, during CT, symptoms of anxiety and depression are higher after the third treatment cycle [8], probably due to the increased side effects. Symptoms tend to increase again at the end of treatment, possibly due to concerns about treatment efficacy, re-entry into daily life, and fear of recurrence [8,13], influencing psychosocial functioning and QoL outcomes [14].

During BC treatment, the concomitance between anxiety and depression symptoms is also frequent [10], as corroborated by the Clark and Watson’s tripartite model [15]. The literature also highlights the high correlation between anxiety and depression as a strong predictor of QoL [9,10,16].

Throughout CT, QoL tends to decrease as treatment cycles increase [17]. This decrease in QoL may persist over a long period (up to 10 years after diagnosis) [18]. During this time, women are also exposed to a high chemical burden as a result of cancer treatment medication, as well as from previous medication to control comorbidities (e.g., diabetes, hypertension, and dyslipidemia). In fact, polypharmacy is very prevalent among cancer patients [19], and, despite being a necessary resource, it may entail risks [20] and worsen long-term health outcomes [21]. In this sense, the use of harmless and natural complementary therapies may be extremely important for mitigating the chemical burden. Essential oils have been used for thousands of years and are reported to provide a variety of benefits, especially for physical and psychological well-being [22], as evidenced by some clinical studies [23,24,25]. In fact, the olfactory cortex is the only sensory system that has a direct relationship with the limbic system, which plays an important role in emotional processing [26].

In a previous study (Pereira et al., submitted), a negative bidirectional temporal relationship between anxiety and QoL, and a positive unidirectional temporal relationship between depression and anxiety were found. According to Lazarus and Folkman’s [27] transactional model of stress and coping, variables can be recursive, i.e., antecedent variables can assume themselves as outcomes, and vice versa. This recursion was also found in a recent study with diabetic patients, which found a negative bidirectional relationship between anxiety, depression, and QoL [28]. These types of prospective analyses regarding these outcomes throughout BC treatment are rare [29], emphasizing its clinical relevance. Likewise, the model of Lazarus and Folkman [27] explains how psychosocial functioning, psychological well-being, and physical health develop over the long term in the presence of stress, but it does not address how these outcomes influence each other over time. However, the authors mention there may be direct relationships between the long-term results [11].

The present study has the following objectives: (1) to assess the temporal relationship between anxiety, depression, and QoL, comparing the control and the experimental group; (2) to assess whether there is an impact of essential oils on this longitudinal relationship; and (3) to outline a hypothetical theoretical model that illustrates the action of hedonic aroma on anxiety, depression, and QoL, based on Lazarus and Folkman’s transactional model of stress and coping. The answers to these objectives can help health professionals understand how relationships between these variables unfold over time, with the aim of intervening as earlier as possible, thus suppressing the long-term relational cascade.

## 2. Methods

### 2.1. Study Design

This is a secondary analysis of a previously reported randomized controlled trial (RCT) (ClinicalTrials.gov (accessed on 12 July 2018), registration: NCT03585218) [30] conducted in three hospitals in the northern region of Portugal between February 2017 and April 2019.

This study was approved by the ethics committees of the three hospitals where the data were collected (Approvals: ref. 39/2017; nº9/2016; CESHB 015/2016). Informed consent was provided by all of the participants. All study procedures were conducted in accordance with the Declaration of Helsinki [31]. There was no change in methods after the start of the intervention.

### 2.2. Sample Size, Participants, and Randomization

The sample size calculation was performed using norms for RCT studies. The difference of 10 points between the groups with a population standard deviation of 15.43 (according to previous study) alpha level of 0.05, effect size of 0.65, and power of 90%, were assumed for the primary outcome, namely QoL scales. Thus, a total of 100 participants (50 participants per group) had to be included.

Eligible participants were women ≥ 18 years old with BC stage I and II who were level 0–2 on the Eastern Cooperative Oncology Group (ECOG) performance status, and who had adjuvant CT scheduled. Participants with psychiatric illness or a cognitive deficit, as reported in their medical records; compromised olfactory function (cut-off = 6 through smell diskettes); and with a low health literacy were excluded.

Of the 134 eligible patients, 19 were excluded, while 115 patients were included and randomized (intervention group: n = 57; control group: n = 58) (see Figure 1). Participants were randomly assigned to the intervention or control group through an allocation ratio of 1:1, using sealed envelopes and single blinding (only researchers had this information). After randomization, balanced conditions between the two groups at baseline were ensured (see Table 1). The study ended up with a total of 112 participants, who were equally distributed between the two groups. This study has a non-probabilistic sample, consisting of 56 participants from the control group (CG) and 56 participants from the experimental group (EG).

### 2.3. Intervention

Bergamot (*Citrus bergamia*), geranium (*Pelargonium graveolens*), and mountain pepper (*Litsea cubeba*) essential oils were used because of their positive effects on anxiety and depression, as well as for their safety and lack of reported side-effects [22]. The essential oils were derived from pure natural and certified plants, cultivated in Italy (bergamot), Egypt (geranium), and China (mountain pepper). Three drops were used in each cotton roll, and were renewed every 30 min throughout the 2 h CT session.

The intervention started during the second cycle of CT (T1 moment), where patients in EG, in addition to the standard CT, inhaled a hedonic aroma that the patient herself selected from three types of essential oils (bergamot, mountain pepper, or geranium). In each session, patients selected and used a single oil, but they could select different oils (out of the three mentioned) in the different sessions. This self-selection constitutes an important part of the positive effects of essential oils [32]. In turn, patients in CG were only treated with standard CT.

The participants of EG were asked, while they received CT, to occasionally (5 in 5 min, as a reference) inhale the selected essential oil until the end of the session (about 2 h) under occasional monitoring by the researcher. In order to prevent patients in EG from contacting the researcher more often, which could bias the results, the researcher made contact with the patients in both groups four times per session.

The intervention occurred during three CT sessions (second, third, and fourth cycles). For inhalation, because the more directly the odoriferous molecules are applied to the nose, the greater their impact on the body [32], it was decided to use cotton rolls saturated with three drops of the oil selected by the participant.

### 2.4. Measurements

For this secondary analysis, the focus was on anxiety, depression, and QoL, which were described as primary outcomes of the main RCT [30].

The European Organization for Research and Treatment of Cancer Quality of Life Questionnaire (EORTC QLQ-C30) [33,34] assesses QoL in cancer patients through 30 items, divided into five functional scales (physical, social, emotional, cognitive, and role), three symptom scales (fatigue, pain, and nausea/vomiting), a global health and QoL scale, and six single item scales (dyspnea, insomnia, appetite, constipation, diarrhea, and financial difficulties). Responses to the items are scored on a four-point scale, where higher scores represent a better QoL. In this study, only the global scale was used with a Cronbach’s alpha of 0.91.

The Hospital Anxiety and Depression Scale (HADS) [35,36] includes 14 items and two subscales: anxiety and depression. Each subscale has seven items, which are scored on a four-point Likert scale (0–3). A high score in each subscale indicates more anxiety and depression. In this study, the two subscales were used. Cronbach’s alpha in the Portuguese version was 0.76 for the anxiety subscale and 0.82 for the depression subscale. In this study, the Cronbach’s alpha was 0.88 for anxiety and 0.91 for depression.

### 2.5. Other Pre-Specified Outcome Measures

Prior to randomization, the following data were collected: patients’ sociodemographic and clinical information (e.g., age, sex, education level, marital status, occupation, type of surgery, disease stage, number of planned cycles, tumor grade, sentinel lymph node, and molecular markers), functional performance assessed using the ECOG performance status (to assist in the inclusion criteria) [37], screening of olfaction using smell diskettes [38], and patient’s literacy assessed using the Short Assessment of Health Literacy [39].

### 2.6. Data Collection Procedure

After the oncology consultation, eligible candidates who agreed to participate in this study were enrolled in both arms. Assessments consisting of self-report questionnaires took place at three points in time: at the second cycle of CT (T1), at the fourth cycle of CT (T3), and three months after the end of CT (T4; post-intervention). Specifically, the baseline moment (T1) corresponds to the beginning of the intervention with the oils. The baseline variables were evaluated before the beginning of the CT in order to assess the impact of the first cycle without the intervention of the oils (see Figure 2).

### 2.7. Data Analysis

Statistical analyses were performed using the R statistical environment (Rstudio, version 3.6.2, R Core Team, 2019, Vienna, Austria) [40], with packages “lavaan” [41] and “semTools” [42]. Autoregressive and cross-lagged structural equation modelling (SEM) was employed to determine the time-related associations between anxiety, depression, and QoL. SEM methodology is a simple statistical tool that allows for analyzing multiple relationships simultaneously.

A three-wave autoregressive cross-lagged model with three variables (anxiety, depressions, and QoL) was specified. The objective was to understand whether the experimental and control groups presented different significant time-related connections in this model. Thus, the theoretical model was assessed for each group, independently. Notice that more sophisticated statistical modelling tools could be used in the assessment of further between-group differences; however, for the purposes of this article, this simpler analysis was preferred, as it avoids complex results and has direct interpretations. For the sake of simplicity, all variables were considered as a single measured variable, in a total parceling form (sum). To compare the relative strengths of the multiple relationships in such models, standardized estimates were used. As a result of attrition and data deviations from normality, the bootstrap standard errors were computed. Figure 3 outlines the theoretical model and its parameter estimation for each group, separately.

## 3. Results

### 3.1. Sociodemographic and Clinical Characteristics

The sociodemographic and clinical characteristics of the sample (n = 112) are shown in Table 1. Concerning these characteristics, there were no significant differences between the intervention group and the control group (*p* < 0.05).

### 3.2. Longitudinal Predictions between Anxiety, Depression, and QoL

The examination of the parameters shows that anxiety at T1 is a temporal predictor of anxiety at T3 ($\beta_{anx,CG}$ = 0.42, 95% CI = [0.01, 0.81]; $\beta_{anx,EG}$ = 0.81, 95% CI = [0.45, 1.00]) in both groups (CG and EG). Thus, higher scores in anxiety at T1 were associated with higher scores of anxiety during CT (T3). A similar situation occurred for QoL ($\beta_{QoL,CG}$ = 0.54, 95% CI = [0.18, 0.89]; $\beta_{QoL,EG}$ = 0.74, 95% CI = [0.48, 0.97]), with worse scores of QoL at T1 being associated with worse scores of QoL during CT (T3).

For EG, QoL at T3 was also a temporal predictor of QoL at T4 (*β* = 0.44, 95% CI = [0.02, 0.81]). Therefore, in this group, worse scores of QoL at T3 were associated with worse scores of QoL three months after the end of CT (T4). Finally, for CG, anxiety at T3 was able to predict depression and QoL ($\beta_{dep}$ = 0.63, 95% CI = [0.06, 1.00], $\beta_{QoL}$ = −0.49, 95% CI = [−0.97, −0.02]) in future time points, more specifically, at T4. Thus, in this group, higher anxiety scores at T3 were associated with higher depression scores and worse QoL scores three months after the end of treatment (T4). Therefore, for women who underwent standard CT treatment, anxiety at T1 was able to predict anxiety six weeks later (T3), which in turn was able to predict future depression and QoL three months after the end of the treatment. In this regard, anxiety acquires a primary role as a long-term temporal precursor.

The autoregressive coefficients of these models allowed for evaluating stability across time. In fact, higher coefficients indicate a higher stability or more influence from previous time points. In these cross-lagged models, in general, stability decreased over time for all of the variables and for both groups (this means that the influence from previous time points became weaker over time). The highest stability was observed for anxiety and QoL in EG during the essential oil intervention (from T1 to T3).

## 4. Discussion

This study aimed to evaluate the longitudinal relationship between anxiety, depression, and QoL in the CG and EG patients. Moreover, the purpose was also to assess whether there is an impact of essential oils on this temporal relationship by comparing the groups. There were no significant differences between the two groups at baseline regarding all sociodemographic and clinical variables. Moreover, both groups underwent exactly the same procedures throughout their chemotherapy treatment, and the only (at least detectable) difference was the oil intervention in the experimental group. Therefore, one may assume that any between-group difference in the outcomes (anxiety, depression, and QoL) over time may be due to the oil intervention.

Overall, the secondary analysis of a previously reported RCT, performed through a cross-lagged model, demonstrated temporal differences between groups, in relation to the primary outcomes of that previous study, i.e., anxiety, depression, and QoL. In CG, it was found that anxiety at T1 was a temporal precursor to anxiety at T3 and that this was a strong predictor of depression and QoL at T4, which was expected according to the literature [16]. The importance of anxiety as a major precursor to depression and QoL in later time points was expected, as it was the most prevalent symptom during CT, especially during the first cycles [8]. In EG, this precursor role of anxiety for long-term depression and QoL was not found. This finding seems to be justified by the greater stability (influence from the previous time points) regarding anxiety in EG, which in turn may be attributed to the oil intervention. This fact was corroborated by other studies reporting that essential oils can improve anxiety [25]. Considering that anxiety is an independent factor for predicting BC recurrence and survival [43], the results are clinically important as they highlight the importance of early intervention in anxiety. In fact, anxiety tends to increase in the first treatment cycles (e.g., [8]), and so it is important to promote stability regarding anxiety early on as a way to mitigate long-term depression and reduced QoL.

The results seem to emphasize the buffer role of essential oils [30], which may affect the adjustment to the disease trajectory [27] through the main precursor of this relationship.

In this study it was also found that the influence anxiety, depression, and QoL at previous times decreased over time. However, the greatest stability across time was found in EG, for anxiety and QoL during the intervention (from T1 to T3). The last result reinforces the action of essential oils as providing temporal stability on anxiety and QoL. However, from T3 to T4, as there was only one intervention session and the time gap was longer (about 5 months, on average), EG also lost stability during that period of time. Thus, the results seem to suggest that if the hedonic aroma intervention was longer, stability regarding anxiety and QoL from T3 to T4 would probably be greater.

Likewise, this study revealed how the outcome variables influenced each other over time, corroborating the direct relationships highlighted by Lazarus and Folkman [27]. However, the influence of variables over time occurred in a unidirectional way, and there were no reciprocal relationships between the variables. Nevertheless, we believe that this interpretation needs some caution, as, if there were additional assessment moments, this recursiveness could eventually become evident.

As a way to explain the theoretical foundations that can support the results of statistical differences between the groups, we hypothesized an explanatory model based on the symbiosis between Lazarus and Folkman’s [27] model and neuroscience fundamentals (e.g., [44,45,46,47]). Thus, we outlined two theoretical models: a model for CG (see Figure 4) and a model for EG (see Figure 5). The two hypothesized models are similar, except for the essential oil inhalation input in EG, which acted directly on women’s life stressors at a very early stage, when stressors were collected by the sensory system. 

In this sense, psychological stress, arising from various sources, triggers a complexity of events in the central nervous system (CNS). This is represented in this model, based on the literature [45,46], through four relational stages, namely: Stage I, characterized by the input of sensory stimuli from the surrounding environment and the respective processing of this information; Stage II, which includes cognitive appraisal and stress responses (emotional and coping), resulting from the prefrontal and limbic system interaction; Stage III, which consists of the initiation of autonomic and endocrine responses through the hypothalamus and brainstem, as well as cognitive reappraisal (cortex and limbic system); and Stage IV, which includes the peripheral nervous system with the body expression of emotions and stress reactions.

In fact, in Stage I, individuals are continuously exposed to various sources of stress in their surrounding environment [48]. Environmental variables influence the way we behave and feel [49]. External sensory information is transmitted from the sensory organs (eyes, nose, skin, ears, and tongue—peripheral nervous system) to the thalamus (central nervous system) [46]. Subsequently, it is relayed by two pathways: the thalamo−amygdala’s pathway and the thalamo−cortico−amygdala’s pathway [47]. The responses of the amygdala to these inputs may be innately programmed or acquired through conditioning, as a result of a learning process over time [50] without the involvement of higher functions. Thus, the amygdala can send signals, through its projections, to the brainstem and hypothalamus [51] and can elicit autonomic and endocrine responses without being subjected to a more elaborate and conscious evaluation [47], as advocated in the model of Lazarus and Folkman [27].

Likewise, and more slowly, the thalamo−cortico−amygdala’s pathway with reciprocal projections arises, in which the amygdala, together with the hippocampus, can receive various inputs from areas of the thalamus; cortical areas; including polysensory areas [47]; and the prefrontal cortex, which will elicit greater cognition (including working memory and decision-making) and awareness of the primary and secondary appraisal, advocated by Lazarus and Folkman [11]. However, purely cognitive processes are emotionally further elaborated through the convergence of the amygdala and hippocampus with areas such as the bed nucleus of the stria terminalis, nucleus accumbens, medial prefrontal cortex, and anterior cingulate gyrus [46,52]. At the so-called limbic−prefrontal junction, external events are consciously evaluated with emotional and motivational meaning, triggering sentimental states, and the initiation of behavioral coping [44]. The outcomes, stemming from Stage II, interact with Stage III, namely, with the hypothalamus acting at the level of the hypothalamic−pituitary−adrenal axis (HPA) with a slower response, and at the sympathetic nervous system (SNS) with a faster response [47]. There may also be a parallel cognitive reappraisal at Stage III [11] to regulate the person’s emotional state, based on the initial assessment of the event and the coping strategies developed. Thus, this reappraisal can change the initial evaluations and the affective states resulting from them [11]. Finally, in Stage IV, the outputs will be transmitted, descending the pathways from the brainstem to the muscles and viscera of our bodies. This transaction includes structures such as the paragigantocellular nucleus, the nucleus of the solitary tract, the intermediolateral cell column, and the pituitary and adrenal glands [44,46].

Responses to psychological and physiological stress are generally considered adaptive, as the activation of Stages III and IV prepares the individual to survive a threat [53]. However, there are contexts in which individuals are submerged in prolonged stress exposure, negatively affecting health outcomes during illness due to their heterogeneous and long-lasting trajectory. The process of physiological overload due to chronic reactivity, called allostatic load, translates into difficulties in adapting to psychological stress [53].

Considering that the emotional responses formed in Stage II determine the results of Stage III and, ultimately, the body’s response to stress (stage IV), the intervention with essential oils sought to act on Stages I and II in order to affect and alter the emotional response of women with BC, with implications for physiological reactivity and health [45]. Based on these premises, a new and neutral stimulus was introduced in EG, namely essential oils, along with external sources of stress, to affect Stage I of the model. As shown in the model hypothesized for EG, olfactory information is transmitted directly to the amygdala, without projecting to the thalamus, unlike the other sensory stimuli [26]. Thus, olfactory stimuli act directly on affective states, without necessarily passing through the sieve of cognitive appraisal, through pathways that allow for purer and less diminished signal processing [26]. Therefore, according to the results of the present study, we can hypothesize that a simple hedonic essential oil may have elicited changes in emotional responses (Stage II), which in turn elicited physiological reactions (stage III), interrupting the internal cascade of cognitive, emotional, and physiological events expected in the face of a stressor such as BC and its treatment. This hypothesis is reinforced by the results found in CG, which corroborate the psychoneuroimmunology (PNI) paradigm [54] and integrate a system of interactions between psycho-behavioral and neuroendocrine-immunological processes that affect the dynamics of health that are sustained in continuous stress−disease relationships [54]. Therefore, by altering affective states and thus behavior (e.g., coping), a more positive cognitive reappraisal of the event (e.g., treatment) may be promoted. However, the effects of essential oils can also be subject to cognitive appraisal through a slower pathway [48], interfering with the symbiosis between cognition and emotion guided in primary and secondary appraisal. Thus, in light of the hypothesized model, future studies should focus on the analysis of cognitive appraisal in order to understand at which moment it is more important to intervene throughout the disease trajectory, as well as its impact on emotional responses and psychosocial adaptation over time. Future studies should also focus on evaluating the impact of essential oils in a long-term cognitive assessment process in order to promote a more positive cognitive reappraisal.

## 5. Limitations and Conclusions

The current study has several strengths, including its randomized design, self-selection of essential oils, reduced dropouts, and long-term (an average of 9 months after diagnosis) follow-up data; however, some limitations need to be acknowledged. Firstly, data were obtained through self-report measures, which may be influenced by social desirability factors. Secondly, direct pathways between anxiety, depression, and QoL were tested over time, without considering other possible sociodemographic and clinical mediators/covariates, such as disease duration, number of cycles, and age, which could interfere with these pathways. Finally, the sample included Stages 1 and 2 of the disease and mostly patients who received breast-conserving surgery; therefore, the results cannot be generalized to all BC patients, thus requiring caution in their interpretation.

This study is one of the first in the literature to examine the temporal reciprocal relationships between anxiety, depression, and QoL in women with early-stage BC, suggesting that, in a standard CT, anxiety during the third cycle of treatment is the main temporal precursor to anxiety throughout the entire treatment, which in turn is the main temporal predictor of depression and QoL three months after CT. Moreover, during the intervention period (from T1 to T3), women who inhaled essential oils showed higher stability over time regarding anxiety and QoL, which is in line with previous results found in the RCT study [30]. Thus, the results show that essential oils may complement chemotherapy in women treated for early-stage BC as a way to improve their long-term adjustment. However, more research is needed to corroborate the mechanisms of the effects hypothesized in this study and to explore how they can be further maximized and better perpetuated over time, probably with a longer intervention program, that is, with a higher number of oil inhalation sessions.

## Figures and Tables

**Figure 1 ijerph-19-09260-f001:**
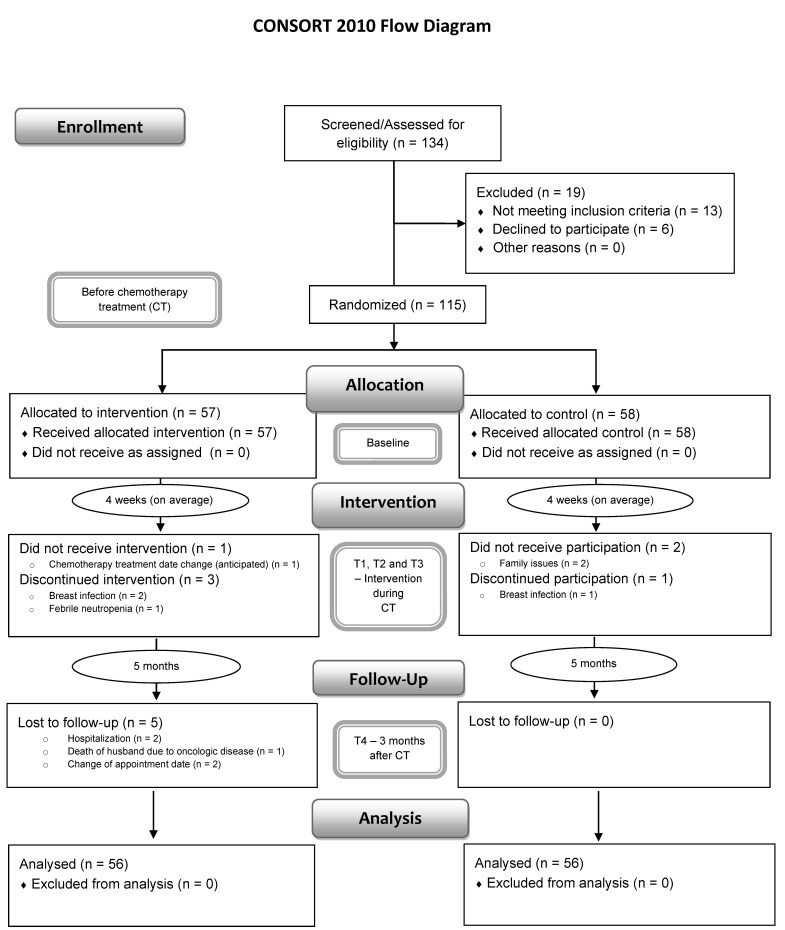
CONSORT flowchart of the present study.

**Figure 2 ijerph-19-09260-f002:**
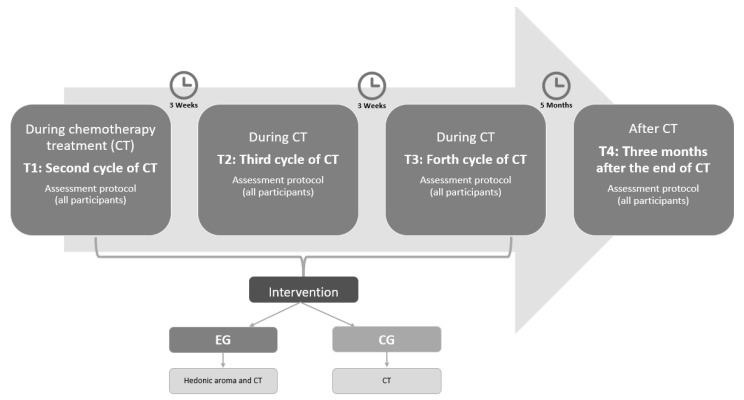
Intervention schedule.

**Figure 3 ijerph-19-09260-f003:**
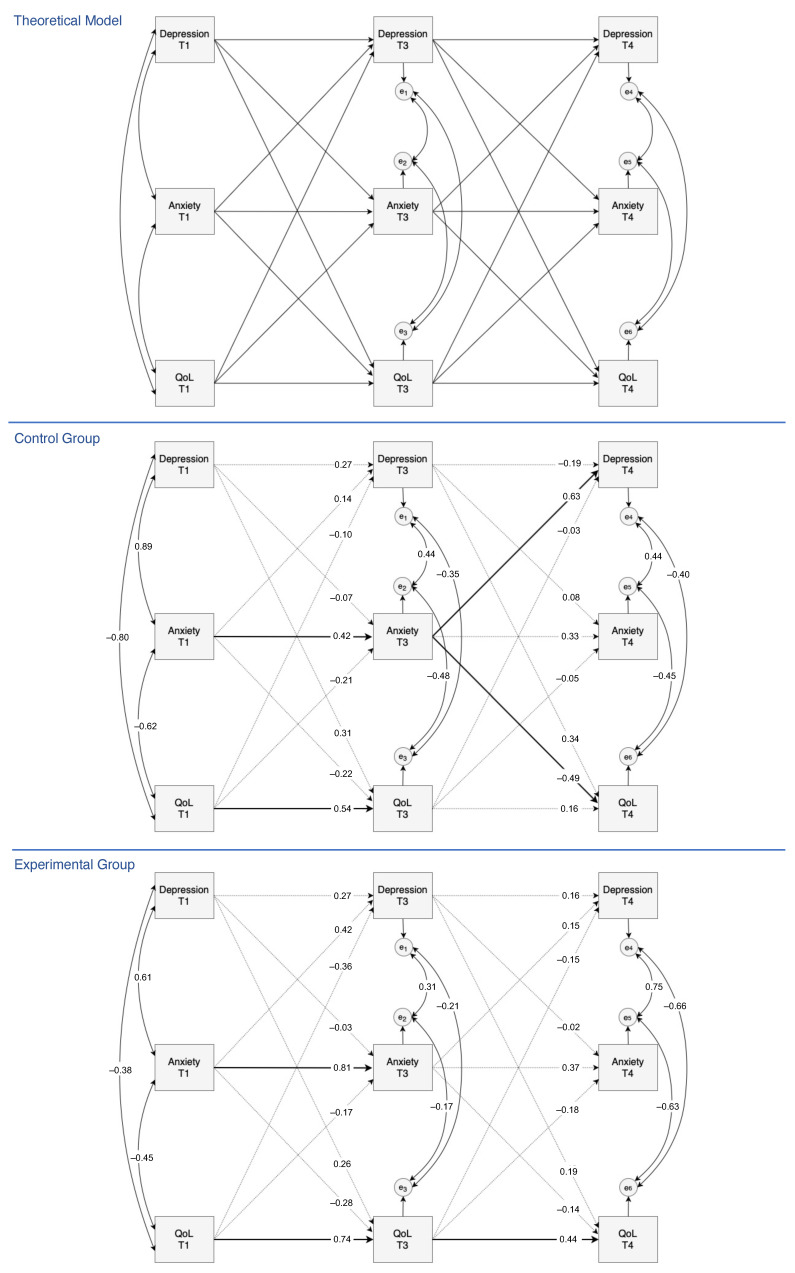
Three-wave structural equation theoretical model with autoregressive and cross-lagged effects for the longitudinal relationship between anxiety, depression, and QoL (up). Below, the model estimation for the control group and for the experimental group. Solid lines represent significant relationships and dotted lines represent no significant relationships.

**Figure 4 ijerph-19-09260-f004:**
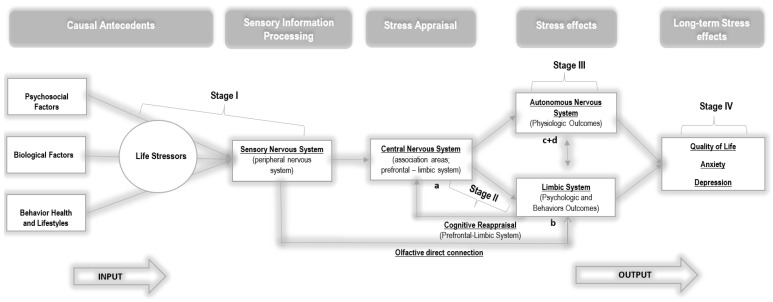
Neuro-psycho-physiological model of stress and coping for the control group. Adapted Transactional Model of Stress and Coping by Lazarus and Folkman [27]. (**a**) Appraisal process arising from interactions between the prefrontal cortex and amygdala, (**b**) the feeling/affect resulting from this appraisal and initiation of behavioral coping, and (**c**) autonomic and endocrine outputs from the hypothalamus, along with (**d**) downward signals to the brainstem and the spinal cord [44].

**Figure 5 ijerph-19-09260-f005:**
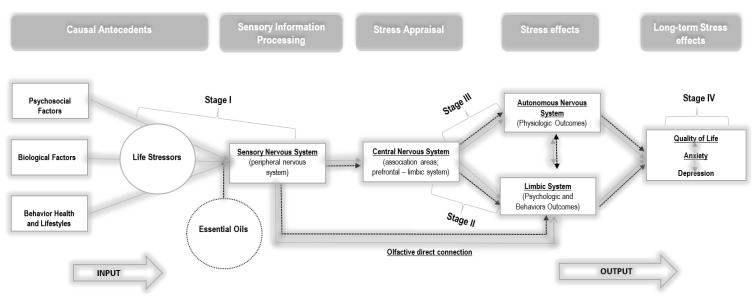
Neuro-psycho-physiological model of stress and coping for the experimental group. Adapted Transactional Model of Stress and Coping by Lazarus and Folkman [27].

**Table 1 ijerph-19-09260-t001:** Socio-demographic and clinical characteristics and balance per group.

T1 (n = 112)	CG (n = 56)	EG (n = 56)	Differences between Groups ^d^
	Freq./M (SD)	Freq./M (SD)	Estimate, *p*-Value
Sociodemographic variables			
Age	51.48 (10.34)	53.50 (10.22)	2.02, *p* = 0.295
Local residence:			
Urban	18	19	−0.08, *p* = 0.841
Rural	38	37	
Marital status:			
Married/cohabiting	45	43	0.213, *p* = 0.645
Unmarried ^a^	11	13	
Education:			
Primary Studies	40	32	0.72, *p* = 0.071
Secondary studies	9	13	
University degree	7	11	
Professional Situation:			
Active	1	3	−1.14, *p* = 0.332
Not active	55	53	
Clinical variables			
Surgery type:			
Lumpectomy	45	44	0.11, *p* = 0.815
Mastectomy ^b^	11	12	
Cancer Stage:			
T1	27	17	0.759, *p* = 0.055
T2	29	39	
Axillary lymph node dissection:			
Yes	29	25	0.29, *p* = 0.450
No	27	31	
Number of Chemotherapy Cycles/Cytotoxic Drugs ^c^			
4 cycles (AC)	18	18	0.36, *p* = 0.297
6 cycles (FEC-D)	15	9	
8 cycles (AC-D)	7	4	
16 cycles (AC-P)	16	25	
Breast Cancer Grade:			
1	7	6	−0.01, *p* = 0.981
2	35	37	
3	14	13	
Months since diagnosis:	2.79 (1.12)	2.96 (1.32)	0.06, *p* = 0.429
Psychological Variables			
QoL	71.45 (16.08)	74.4 (14.79)	0.13, *p* = 0.356
Anxiety	6.91 (4.09)	5.25 (3.53)	−0.28, *p* = 0.076
Depression	4.44 (3.63)	3.93 (2.94)	−0.11, *p* = 0.504

Note: M = mean; SD = standard deviation; Freq. = frequencies. ^a^ Includes single/separated/widowed/divorced. ^b^ Includes modified radical mastectomy, single mastectomy, and bilateral mastectomy. ^c^ AC = adriamycin−cyclophosphamide; FEC-D = 5-fluorouracil/epirubicin/cyclophosphamide followed by docetaxel; AC-D = adriamycin−cyclophosphamide followed by docetaxel; AC-P = adriamycin−cyclophosphamide followed by paclitaxel. ^d^ Estimated differences and significances were assessed through regression modelling. Appropriate regressions (with default links) were selected according to the dependent variable: normal (age), logistic (local residence, marital status, professional situation, surgery type, BC stage, and axillary lymph node dissection), ordinal (education, CT cycles, and BC grade), Conway−Maxwell−Poisson (months since diagnosis), and beta (QoL, anxiety, and depression).

## Data Availability

The data used to support the findings of this study are available from the corresponding author upon request.

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
