# Peer review of "How Does Hedonic Aroma Impact Long-Term Anxiety, Depression, and Quality of Life in Women with Breast Cancer? A Cross-Lagged Panel Model Analysis"

_ijerph, 2022, doi:10.3390/ijerph19159260_

Round 1

Reviewer 1 Report

This report is considered as an interesting study to confirm the predictors of anxiety, depression, and QOL by chemotherapy period. In addition, the hypotheses and results of the effects of aromatic oils in the course of this treatment are very interesting.

However, we believe that cancer patients' anxiety, depression and QOL scores are related to the patient's symptoms (pain, etc.) or prognosis after chemotherapy.

Without excluding other factors that may affect anxiety, depression, and QOL scores in breast cancer patients, it would be a hasty conclusion to determine the effects of aroma oil. This paper seems to need further explanation on this point.

Author Response

Comments and Suggestions for Authors

This report is considered as an interesting study to confirm the predictors of anxiety, depression, and QOL by chemotherapy period. In addition, the hypotheses and results of the effects of aromatic oils in the course of this treatment are very interesting.

However, we believe that cancer patients' anxiety, depression and QOL scores are related to the patient's symptoms (pain, etc.) or prognosis after chemotherapy.

Without excluding other factors that may affect anxiety, depression, and QOL scores in breast cancer patients, it would be a hasty conclusion to determine the effects of aroma oil. This paper seems to need further explanation on this point.

Answer: This information was clarified in the discussion (please, see lines 264-269).

Author Response

For Authors,

This study examines the long-term effects on psychological indicators regarding the introduction of essential oils into the care of breast cancer patients. The reviewer thought the paper was useful for readers and attached several suggestions. I hope it will be helpful for revising your manuscript.

Throughout, there were some parts that were redundant and difficult to read, so an English review is recommended.

P5 L146: Intervention

  1. Regarding the selection of essential oils by the patient, once the decision was made at the beginning, did you use the same ones until the end?

Answer: In each session, patients selected and used a single oil, but they could select different oils (out of three) in different sessions. According to Pavlov's principle of classical conditioning and in order to alleviate the conditioning of a neutral stimulus (essential oils), in each chemotherapy session, the participant should always choose the oil, being able to change from session to session.

The research team just wanted to guarantee that patients inhaled a non-neurotoxic and anxiolytic oil (the three oils had these properties).

This information was added in the manuscript (lines 160-161).

  1. Does salivary roll (L151-152) refer to cotton roll?

Answer: Yes, cotton roll. We changed the manuscript accordingly (lines 155).

  1. In presenting the essential oils, did you stick the cotton roll under the patient's nose? Please describe more carefully.

Answer: No. The participant was asked to occasionally inhale the cotton roll, in order to prevent saturation of the olfactory receptors. We monitored throughout the chemotherapy session (every 30 minutes) if they were using the cotton roll with the oil. This information was added (lines 164-165).

P6 L93: Data collection procedure

4.The experimental schedule is a little confusing; please make it easier for readers to understand by making better use of Figure 1 or creating a separate figure.

Answer: We created and added a new figure to clarify the experimental timeline. (figure 2) (line 206).

P10 Fig.3,4

5.What is CG's olfactive direct connection? Also, was Cognitive Reappraisal not done in EG?

Answer: The direct olfactory connection in the CG is the same as in the EG, since it is something structural anatomically. The difference is that in the CG it was not activated, because the participants did not use the essential oils, and in the experimental group was activated. For this reason, there is an enhancement of the arrow, highlighting the direct action to the limbic system and in this way the quick action in emotional terms. The GC mirrors well-known mechanisms from the literature, and the EG intends to show the action and pathway of essential oils in neuroanatomical terms.

Cognitive reappraisal was not evaluated, it was only hypothesized in the discussion, based on Lazarus and Folkman's transactional model. Nevertheless, the importance of studying this pathway, in upcoming studies, is emphasized.

Reviewer 3 Report

The manuscript entitled: „ How Does a Hedonic Aroma Impact Long-Term Anxiety, Depression, and Quality of Life in Women with Breast Cancer? A Cross-Lagged Panel Model Analysis.” is an interesting, properly designed, well-written and comprehensive study. The Introduction and Discussion sections are very supportive in understanding the results and conclusions. It is classified as an original article and meets the scope of the journal.

1. 1. Breast cancer (BC) is not only a leading female malignancy in developed countries but also a rising trend in  BC incidence in these populations are observed that additionally supports new research on BC – Authors should consider the above information in Introduction – the following reference might be helpful:

Wojtyla C, Bertuccio P, Wojtyla A, La Vecchia C. European trends in breast cancer mortality, 1980–2017 and predictions to 2025. Eur J Cancer. 2021;152:4–17.

·        Piechocki M, KozioÅ‚ek W, Sroka D, Matrejek A, MizioÅ‚ek P, Saiuk N, Sledzik M, Jaworska A, Bereza K, Pluta E, Banas T. Trends in Incidence and Mortality of Gynecological and Breast Cancers in Poland (1980-2018). Clin Epidemiol. 2022 Jan 24;14:95-114. doi: 10.2147/CLEP.S330081. PMID: 35115839; PMCID: PMC8800373.

      Carioli G, Malvezzi M, Rodriguez T, Bertuccio P, Negri E, La Vecchia C. Trends and predictions to 2020 in breast cancer mortality in Europe. The Breast. 2017;36:89–95. doi:10.1016/j.breast.2017.06.003

·        

2.      2. Introduction: “Breast cancer (BC), the leading cancer disease in women worldwide (Ferlay et al., 392019), is a heterogeneous disease, with different treatments: systemic therapy (chemotherapy and hormone therapy) and local therapy (surgery and radiotherapy) used to improve patient survival outcomes (Cardoso et al., 2019).” – please also include immunotherapy as an important therapeutic option.

3.      3. Methods: “Of the 134 eligible patients, 19 were excluded from participating in this study, while 115 patients were eligible, included, and randomized (intervention group: n = 57; control 134 group: n = 58).” – after this sentence please redirect readers ti the Figure 1

4.      4. Results: “The examination of the parameters showed that anxiety at T1 is a temporal predictor of anxiety at T3 (…)” – please specify if this chapter the whole group or CG only.

5.      5. The resolution of Figures should be improved.

Author Response

The manuscript entitled: “How Does a Hedonic Aroma Impact Long-Term Anxiety, Depression, and Quality of Life in Women with Breast Cancer? A Cross-Lagged Panel Model Analysis.” is an interesting, properly designed, well-written and comprehensive study. The Introduction and Discussion sections are very supportive in understanding the results and conclusions. It is classified as an original article and meets the scope of the journal.

  1. Breast cancer (BC) is not only a leading female malignancy in developed countries but also a rising trend in BC incidence in these populations are observed that additionally supports new research on BC – Authors should consider the above information in Introduction – the following reference might be helpful:

Wojtyla C, Bertuccio P, Wojtyla A, La Vecchia C. European trends in breast cancer mortality, 1980–2017 and predictions to 2025. Eur J Cancer. 2021;152:4–17.

Piechocki M, Koziołek W, Sroka D, Matrejek A, Miziołek P, Saiuk N, Sledzik M, Jaworska A, Bereza K, Pluta E, Banas T. Trends in Incidence and Mortality of Gynecological and Breast Cancers in Poland (1980-2018). Clin Epidemiol. 2022 Jan 24;14:95-114. doi: 10.2147/CLEP.S330081. PMID: 35115839; PMCID: PMC8800373.

      Carioli G, Malvezzi M, Rodriguez T, Bertuccio P, Negri E, La Vecchia C. Trends and predictions to 2020 in breast cancer mortality in Europe. The Breast. 2017;36:89–95. doi:10.1016/j.breast.2017.06.003

Answer: The authors thank the reviewer for this request. The manuscript was changed accordingly (lines 40-43).

  1. Introduction: “Breast cancer (BC), the leading cancer disease in women worldwide (Ferlay et al., 392019), is a heterogeneous disease, with different treatments: systemic therapy (chemotherapy and hormone therapy) and local therapy (surgery and radiotherapy) used to improve patient survival outcomes (Cardoso et al., 2019).” – please also include immunotherapy as an important therapeutic option.

Answer: The manuscript was changed accordingly (line 44).

  1. Methods: “Of the 134 eligible patients, 19 were excluded from participating in this study, while 115 patients were eligible, included, and randomized (intervention group: n = 57; control 134 group: n = 58).” – after this sentence please redirect readers ti the Figure 1

Answer: The manuscript was changed accordingly (line 138).

  1. Results: “The examination of the parameters showed that anxiety at T1 is a temporal predictor of anxiety at T3 (…)” – please specify if this chapter the whole group or CG only.

Answer: This sentence was clarified (lines 234-235).

  1. The resolution of Figures should be improved.

Answer: Figures resolution was improved.